# Theoretical guarantees for the EM algorithm when applied to mis-specified Gaussian mixture models

**Raaz Dwivedi**[*]   **Nhat Ho**[*]   **Koulik Khamaru**[*]
UC Berkeley
{raaz.rsk, minhnhat, koulik}@berkeley.edu

**Martin J. Wainwright**
UC Berkeley
Voleon Group
wainwrig@berkeley.edu

**Michael I. Jordan**
UC Berkeley
jordan@berkeley.edu

## Abstract

Recent years have witnessed substantial progress in understanding the behavior of EM for mixture models that are correctly specified. Given that model mis-specification is common in practice, it is important to understand EM in this more general setting. We provide non-asymptotic guarantees for the population and sample-based EM algorithms when used to estimate parameters of certain mis-specified Gaussian mixture models. Due to mis-specification, the EM iterates no longer converge to the true model and instead converge to the projection of the true model onto the fitted model class. We provide two classes of theoretical guarantees: (a) a characterization of the bias introduced due to the mis-specification; and (b) guarantees of geometric convergence of the population EM to the model projection given a suitable initialization. This geometric convergence rate for population EM implies that the EM algorithm based on $n$ samples converges to an estimate with $1/\sqrt{n}$ accuracy. We validate our theoretical findings in different cases via several numerical examples.

## 1 Introduction

Mixture models play a central role in statistical applications, where they are used to capture heterogeneity of data arising from several underlying subpopulations. However, estimating the parameters of mixture models is a challenging task, due to the non-convexity of the log likelihood function. As shown by classical work, the maximum likelihood estimate (MLE) often has good properties for mixture models, but its computation can be non-trivial. One of the most popular algorithms used to compute the MLE (approximately) is the expectation maximization (EM) algorithm. Although EM is widely used in practice, it does not always converge to the MLE, and its convergence rate can vary as a function of the problem. Classical results provide guarantees about the convergence rates of EM to local maxima [4, 16]. In the specific setting of Gaussian mixtures, population EM (idealized EM with infinite samples) was shown to have a range of behavior from super-linear convergence to slow convergence like a first-order method depending on the overlap between the mixtures [9, 18]. More recently, there has been a renewed interest in providing explicit and non-asymptotic guarantees on the convergence of EM. Notably, Balakrishnan et al. [1] developed a rather general framework for characterizing the convergence of EM. For well-specified problems—including the two-component Gaussian location mixture as a particular example—they provided sufficient conditions for the EM

---

[*]Raaz Dwivedi, Nhat Ho, and Koulik Khamaru contributed equally to this work.

algorithm to converge to a small neighborhood of global maximum; in addition, they provided explicit bounds on the sample complexity of EM, meaning the number of samples $n$ required, as a function of the tolerance $\epsilon$, problem dimension and other parameters, to achieve an $\epsilon$-accurate solution. A line of follow-up work has generalized and extended results of this type (e.g., see the papers [20, 15, 7, 17, 3, 19, 5, 2]).

A shared assumption common to this body of past work is that either the true distribution of each subpopulations is known, or that the number of components is exactly known; in practice, both of these conditions are often violated. In such settings, it is well known that the MLE, instead of approximating the true parameter, approximates a Kullback-Leibler projection of the data-generating distribution onto the fitted model class. Thus, the MLE exhibits a desirable form of robustness to model mis-specification.

On the other hand, it is not obvious *a priori* that this robustness need be shared by the solutions returned by the EM algorithm. Since these solutions are those actually used in practice, it is important understand under what conditions the EM algorithm, when applied with mis-specified models, converges to an (approximate) KL projection. The main contribution of this paper is to provide some precise answers to this question, and moreover to quantify the bias that arises from model mis-specification. Our analysis focuses on two classes of mis-specified mixture models.

- *Under-specified number of components*: Suppose that the true model is given by location-shifted mixture of $k \geq 3$ univariate Gaussians, but we use EM to fit a location-shifted Gaussian mixture with $k - 1$ components. This scenario is very common: it arises naturally when either the mixture components are very close or some of the mixture weights are very small, so that the data generating distribution appears to have fewer components. Analysis of the EM algorithm when the fitting distribution has fewer mixture components than the data-generating distribution poses new challenges; in particular, it requires an understanding of the *model bias*, meaning the Kullback-Leibler discrepancy between the true model from its projection(s) onto the class of fitted models. In this paper, we provide a detailed analysis of the $k = 3$ case. First, we characterize the model bias induced by fitting a two-component mixture to a three-component mixture with unknown means but known variance. We then provide sufficient conditions for the population EM updates to converge at a geometric rate to the KL projection of the true model onto the fitted model class. Finally, using Rademacher-complexity based arguments and the geometric convergence of population EM, we conclude that with high probability, the EM updates with $n$ samples converge to a ball of radius $1/\sqrt{n}$ around the aforementioned KL projection.

- *Incorrectly specified weights or variances*: In our second problem class, we assume that the number of components is correctly specified, but either the mixture weights or the variances are mis-specified. Concretely, suppose that the true model is a two-component location-shifted Gaussian mixture with weights/variances that differ from those in the fitted model class. Our analysis reveals a rather surprising phenomenon with respect to EM convergence: despite the potential non-convexity of the problem, the iterates converge at a geometric rate to a unique fixed point from an arbitrary initialization. Our results suggest that the projection from the true model to the fitted model is actually unique. Finally, we prove that the sample-based EM updates achieve standard minimax convergence rate of order $1/\sqrt{n}$.

Table 1 provides a high-level summary of our results, where we use $(\theta, \sigma, \alpha)$ to denote the Gaussian mixture component with mean $\theta$, variance $\sigma^2$ and weight $\alpha$, i.e., $\alpha \mathcal{N}(\theta, \sigma^2)$.

The remainder of our paper is organized as follows. In Section 2, we introduce the problem set-up and provide the background information on the EM algorithm. In Section 3, we present our results for the first framework and provide expressions for the bias and rate of convergence of EM for different 3 component mixture of Gaussians. Section 4 contains results when the mixture weights and variance are mis-specified. Numerical experiments illustrating our theoretical results are presented in Section 5. Finally in Section 6, we conclude the paper with a discussion of our results and a few possible venues for future work.

**Notation**: We use $c, c', c_1, c_2$ to denote universal constants whose value may vary in different contexts. For two distributions $\mathbb{P}$ and $\mathbb{Q}$, the Kullback-Leibler divergence between them is denoted by $\text{KL}(\mathbb{P}, \mathbb{Q})$. We use the standard big-O notation to depict the scaling with respect to a particular quantity and hide constants and other problem parameters.

| True Model | Best fit with two components | Bias $\min\{|\bar{\theta} - \theta^*|, |\bar{\theta} + \theta^*|\}$ | Statistical error $|\hat{\theta}_n - \bar{\theta}|$ of sample EM |
|---|---|---|---|
| 3-component mixture: $(-\theta^*(1+\rho), \sigma, 1/4);$ $(-\theta^*(1-\rho), \sigma, 1/4);$ $(\theta^*, \sigma, 1/2)$ | $(-\bar{\theta}, \sigma, 1/2);$ $(\bar{\theta}, \sigma, 1/2)$ $\sigma$ known | $\rho\,|\theta^*| + c\,(\rho\,|\theta^*|/\sigma)^{1/4}$ | $n^{-1/2}$ |
| 3-component mixture: $(-\theta^*, \sigma, (1-\omega)/2);$ $(\theta^*, \sigma, (1-\omega)/2);$ $(0, \sigma^2, \omega)$ | $(-\bar{\theta}, \sigma, 1/2);$ $(\bar{\theta}, \sigma, 1/2);$ $\sigma$ known | $\dfrac{c\omega^{1/8}\,|\theta^*|^{1/4}}{\sqrt{1-\omega}\,\sigma^{1/4}}$ | $n^{-1/2}$ |
| 2-component mixture: $(-\theta^*, \sqrt{\sigma^2 - \theta^{*2}}, 1/2);$ $(\theta^*, \sqrt{\sigma^2 - \theta^{*2}}, 1/2);$ | $(-\bar{\theta}, \sigma, \pi);$ $(\bar{\theta}, \sigma, 1-\pi)$ $\sigma, \pi \neq 1/2$ known | $\dfrac{c\,|\theta^*|\,((2-4\pi)+\theta^{*2})^{1/2}}{\sigma}$ | $n^{-1/2}$ |

**Table 1.** Summary of the main theoretical gurantees of this paper. Here the parameter $\theta^*$ denotes the true parameter value (in the data-generating distribution), $\bar{\theta}$ denotes the value of the parameter of the best fit model, and $\hat{\theta}_n$ denotes the estimate returned by running the EM algorithm. Recall that the true model is not in the class of fitted models, and we can only hope to estimate $\bar{\theta}$; consequently, in the above table lists the performance of the EM algorithm in estimating $\bar{\theta}$ for different settings. The first column lists the true model, while the second column shows the fitted model. In the third column, we summarize the bias of the parameter of the best fitted model (2). When using EM with $n$ samples, the final statistical error $|\hat{\theta}_n - \bar{\theta}|$ has the statistical rate of order $n^{-1/2}$ in all cases, as depicted in the fourth column (here $\hat{\theta}_n$ denotes the final sample EM estimate).

## 2 Problem set-up

Throughout this paper, we assume that data is generated according to some true distribution $\mathbb{P}_*$, which admits a continuous density over $\mathbb{R}$. We are interested in the performance of the EM algorithm when we fit the model below using a two-component mixture of location-shifted Gaussians with known variance $\sigma^2$ and known mixture weight $\pi \in (0, 1)$:

$$\mathbb{P}_\theta = \pi \mathcal{N}(\theta, \sigma^2) + (1 - \pi)\mathcal{N}(-\theta, \sigma^2) \tag{1}$$

We consider two distinct settings of the mixture weights in model (1):

- **Balanced mixtures**: the mixture weights are assumed to be equal, i.e., $\pi = 1 - \pi = 1/2$.

- **Unbalanced mixtures**: the mixture weights are assumed to be unequal $\pi = \frac{1}{2}(1 - \epsilon)$ and $1 - \pi = \frac{1}{2}(1 + \epsilon)$ where $|\epsilon| \in (0, 1)$.

In order to estimate the location parameters, we apply the EM algorithm, allowing $\theta$ to vary over some compact set $\Theta$. Since the true distribution $\mathbb{P}_*$ may not belong to the class of fitted models, the best possible estimator is the projections of $\mathbb{P}_*$ to the fitted model (1). It is given by

$$\bar{\theta} \in \arg\min_{\theta \in \Theta} \mathrm{KL}\left(\mathbb{P}_*, \mathbb{P}_\theta\right). \tag{2}$$

Our main goal in the paper is to establish the convergence rate of EM updates to $\bar{\theta}$ for various choices of the data-generating model $\mathbb{P}_*$ and the fitted model (1).

### 2.1 EM algorithm for two-component location-Gaussian mixtures

Let us now introduce some notation as well as a brief description of the EM algorithm for two-component Gaussian location mixtures (1). The population version of EM is based on the function

$$Q(\theta'; \theta) := -\frac{1}{2}\mathbb{E}\left[w_\theta(X)\left(X - \theta'\right)^2 + (1 - w_\theta(X))\left(X + \theta'\right)^2\right], \tag{3}$$

where the expectation is taken over the true distribution $\mathbb{P}_*$. For any fixed $\theta$, the M-step in the EM updates for the model (1) is obtained by maximizing the minorization function (3); for a detailed

derivation, see the paper [1]. More precisely, we denote the population EM operator $M : \mathbb{R} \to \mathbb{R}$ as

$$M(\theta) := \arg\max_{\theta'} Q(\theta', \theta) = \mathbb{E}\left[(2w_\theta(X) - 1)X\right], \tag{4a}$$

where the weighting function $w_\theta$ in the above formulation is given by

$$w_\theta(x) := \frac{\pi \exp\left(-\frac{(\theta-x)^2}{2\sigma^2}\right)}{\pi \exp\left(-\frac{(\theta-x)^2}{2\sigma^2}\right) + (1 - \pi) \exp\left(-\frac{(\theta+x)^2}{2\sigma^2}\right)}. \tag{4b}$$

Note that the parameter $\bar{\theta}$, defined in equation (2), minimizes the KL-distance between the fitted model and the true model, thereby ensuring that the log-likelihood is maximized at the model indexed by the parameter $\bar{\theta}$. Consequently, the parameter $\bar{\theta}$ is a fixed point of the population EM update—that is, $M(\bar{\theta}) = \bar{\theta}$. The sample version of the EM algorithm—the method actually used in practice—is obtained by simply replacing the expectations in equations (3) and equation (4a) by the sample-based counterpart. In particular, given a set of $n$ i.i.d. samples $\{X_i\}_{i=1}^n$ from the true model, the sample EM operator $M_n : \mathbb{R} \mapsto \mathbb{R}$ takes the form

$$M_n(\theta) := \frac{1}{n} \sum_{i=1}^n (2w_\theta(X_i) - 1)X_i. \tag{5}$$

With this notation in place, we are now ready to state our main results.

## 3   Guarantees for EM algorithm for mis-specified number of components

In this section, we study the convergence of the EM algorithm in the setting of under-fitted mixtures, where the number of components in the true model is larger than that in the fitted model. In sharp contrast to the traditional setting of correctly specified mixture models, where the number of components of the true model is known to the EM algorithm, we analyze the performance of the EM algorithm in the setting where the true number of the components is not known. Such a scenario naturally occur in many practical cases, examples include: (1) Some components in the mixture are very close, and it is hard to distinguish them; (2) Some components have very small mixture weights and thereby are difficult to detect. Consequently, in the aforementioned situations, the number of components observed from the data may be much smaller compared to the number of components present in the true model. In this section, we characterize the bias of the two-component fit and analyze the convergence properties of EM for such a fit.

### 3.1   Three-component mixtures with two close components

First, we consider the case, where the true model has distribution $\mathbb{P}_*$ is a mixture of three-component Gaussian location mixture given by

$$\mathbb{P}_* = \frac{1}{4}\mathcal{N}(-\theta^*(1+\rho), \sigma^2) + \frac{1}{4}\mathcal{N}(-\theta^*(1-\rho), \sigma^2) + \frac{1}{2}\mathcal{N}(\theta^*, \sigma^2) \tag{6}$$

for some $\theta^*$ in a compact subset $\Theta$ of the real line, and a small positive scalar $\rho$ that characterizes the separation between two cluster means $-\theta^*(1+\rho)$ and $-\theta^*(1-\rho)$. For fitting the model, we assume that the variance $\sigma^2$ is known, and we suspect that the true model is a two-component mixture (since $\rho$ is small). Consequently, we fit the data with the model

$$\mathbb{P}_\theta = \frac{1}{2}\mathcal{N}(-\theta, \sigma^2) + \frac{1}{2}\mathcal{N}(\theta, \sigma^2), \tag{7}$$

and we use the EM algorithm to estimate the location parameter $\theta$. Clearly, the performance of model (7) and consequently the EM algorithm depends on the relationship between the separation factor $\rho$ and the SNR $\eta := |\theta^*|/\sigma$ of the true model (6). Since the true model does not belong in the family of two components location-Gaussian mixtures in model class (7), the role of the projection parameter $\bar{\theta} \in \arg\min_{\theta \in \Theta} \mathrm{KL}(\mathbb{P}_*, \mathbb{P}_\theta)$ becomes crucial. In the next proposition, we provide an explicit bound for the bias between $\bar{\theta}$ and $\theta^*$ as a function of the problem parameters.

**Proposition 1.** *Given the true model* (6) *and any* $\rho > 0$*, we have*

$$\min\left\{|\theta^* - \bar{\theta}|, |\theta^* + \bar{\theta}|\right\} \le \rho|\theta^*| + c\left(\frac{\rho|\theta^*|}{\sigma}\right)^{1/4}, \tag{8}$$

*where $c$ is a universal positive constant that depends only on the set $\Theta$.*

In order to simplify our results in the sequel, we assume that $\eta = |\theta^*|/\sigma \ge 1$ and use a simpler bound on the bias—viz.:

$$\min\left\{|\theta^* - \bar{\theta}|, |\theta^* + \bar{\theta}|\right\} \le \left(\rho + \frac{\rho^{1/4}}{\eta^{3/4}\sigma}\right)|\theta^*| \le \left(\rho + \frac{\rho^{1/4}}{\sigma}\right)|\theta^*|. \tag{9}$$

The bound above directly implies that $|\bar{\theta}|$ belongs to the interval $[(1 - C_\rho)|\theta^*|, (1 + C_\rho)|\theta^*|]$, assuming that $C_\rho := \rho + \rho^{1/4}/\sigma \le 1$. As $\rho \to 0$, we have $C_\rho \to 0$ implying that $|\theta^*|$ and $|\bar{\theta}|$ are almost identical. In the sequel, we utilize this precise control of $|\bar{\theta}|$ in terms of $|\theta^*|$, provided by Proposition 1, to analyze the behavior of the EM algorithm in a neighborhood of $\bar{\theta}$. Defining $\rho_\star := \sup\{\rho > 0 | C_\rho \le 1/9\}$, the following result characterizes the behavior population EM operator for the three-component Gaussian location mixture described by equation (6).

**Theorem 1.** *There exist universal constants $c', c''$ such that the population EM operator for model (6) with $\rho \le \rho_\star$ and $\eta \ge c'$ satisfies*

$$\left|M(\theta) - \bar{\theta}\right| = \mathbb{E}\left|2(w_\theta(X) - w_{\bar{\theta}}(X))X\right| \le \gamma\left|\theta - \bar{\theta}\right|, \quad \text{for any } \theta \in \mathbb{B}(\bar{\theta}, |\bar{\theta}|/4).$$

In words, Theorem 1 establishes that the population EM iterates (in the ideal, infinite data limit) are $\gamma$-contractive with respect to $\bar{\theta}$ over the ball $\mathbb{B}(\bar{\theta}, |\bar{\theta}|/4)$, where $\gamma \le e^{-c''\eta^2}$. Combining that result with the condition $C_\rho \le 1/9$, we can demonstrate that $|\bar{\theta}|$ is unique (See Section A.1.4 in the Appendix). These results have a direct implication for the *sample-based version* of EM that is implemented in practice. In particular, the next result shows that EM updates with $n$ samples converge in a constant number of steps to a neighborhood of $\bar{\theta}$.

**Corollary 1.** *Consider any scalar $\delta \in (0, 1)$, sample size $n \ge c_1 \log(1/\delta)$ and starting point $\theta^0 \in \mathbb{B}(\bar{\theta}, |\bar{\theta}|/4)$. Then under the assumptions of Theorem 1, the sample-based EM sequence $\theta^{t+1} = M_n(\theta^t)$ for the model (6) satisfies*

$$\left|\theta^t - \bar{\theta}\right| \le \gamma^t\left|\theta^0 - \bar{\theta}\right| + \frac{c_2}{1 - \gamma}|\theta^*|\left(\theta^{*2} + \sigma^2\right)\sqrt{\frac{\log(1/\delta)}{n}} \tag{10}$$

*with probability at least $1 - \delta$, where $\gamma \le e^{-c'\eta^2}$.*

Note that the bound (10) consists of two main terms: the first term captures the geometric convergence of the population EM operator from Theorem 1, while the second term characterizes the radius of convergence in terms of sample complexity, which is $\mathcal{O}(\sqrt{1/n})$. Therefore, with probability at least $1 - \delta$, we have

$$\left|\theta^T - \bar{\theta}\right| \le \frac{c|\theta^*|(\theta^{*2} + \sigma^2)}{1 - \gamma}\sqrt{\frac{\log(1/\delta)}{n}} \quad \text{for} \quad T \ge c'\frac{\log(n/(\log(1/\delta)|\theta^*|(\theta^{*2} + \sigma^2)))}{\log(1/\gamma)},$$

where $c, c'$ are universal constants.

### 3.2 Three-component mixtures with small weight for one component

Next, we consider the case where the true model $\mathbb{P}_*$ is a three-component Gaussian location mixture model of the form

$$\mathbb{P}_* = \frac{1 - \omega}{2}\mathcal{N}(-\theta^*, \sigma^2) + \omega\mathcal{N}(0, \sigma^2) + \frac{1 - \omega}{2}\mathcal{N}(\theta^*, \sigma^2). \tag{11}$$

In other words, two components are dominant with means $-\theta^*$ and $\theta^*$ respectively, and we have a small component at the origin. For such a model, it is again conceivable to fit a 2-component mixture given by equation (7). The primary interest in such a setting is driven by the fact that, when $\omega > 0$ is sufficiently small, recovering the third small component with center at origin is usually hard; consequently clustering that component with one of the other two may be a good idea. Once again, the convergence of EM is governed by the properties of $\bar{\theta}$ that we characterize in the next proposition.

**Proposition 2.** *For the three components location-Gaussian mixtures in model* (11), *we have*

$$\min\left\{\left|\theta^* - \overline{\theta}\right|, \left|\theta^* + \overline{\theta}\right|\right\} \leq \frac{c\omega^{1/8}\left|\theta^*\right|^{1/4}}{\sigma^{1/4}\sqrt{1-\omega}}, \tag{12}$$

*where c is a universal positive constant that depends only on the set $\Theta$.*

In order to simplify further results, we assume under the condition $\eta := \frac{\theta^*}{\sigma} \geq 1$. Then we have $\min\left\{\left|\theta^* - \overline{\theta}\right|, \left|\theta^* + \overline{\theta}\right|\right\} \leq C_\omega \left|\theta^*\right|$, where $C_\omega := c\omega^{1/8}/(\sigma\sqrt{1-\omega})$. Such a bound on bias leads to slightly different conditions for convergence of the EM algorithm for model (11) compared to the EM convergence for model (6). Note that for any fixed variance $\sigma^2$, the function $C_\omega$ increases with $\omega$ and $C_0 = 0$. Let $\omega_\star = \sup\{\omega > 0 | C(\omega) \leq 1/9\}$. Similar to the model (6), we analyze the convergence rate of EM under a strong SNR condition of true model (11). We define $\tilde{\gamma} := \gamma(\eta, \omega) = (1-\omega)e^{-\eta^2/64} + \omega < 1$. With the above notations in place, we now establish the contraction of the population EM operator $M(\theta)$ for the three components location-Gaussian mixture (11).

**Theorem 2.** *For SNR $\eta \geq 1$ sufficiently large and $\omega \leq \omega_\star$, and for any $\theta^0 \in \mathbb{B}(\overline{\theta}, \left|\overline{\theta}\right|/4)$, the population EM operator for the Gaussian mixture* (11) *satisfies*

$$\left|M(\theta^0) - \overline{\theta}\right| = \mathbb{E}\left|2(w_{\theta^0}(X) - w_{\overline{\theta}}(X))X\right| \leq \tilde{\gamma}\left|\theta^0 - \overline{\theta}\right|. \tag{13}$$

*Consequently, the population EM sequence $\theta^{t+1} = M(\theta^t)$ converges to $\overline{\theta}$ at a linear rate.*

The precise expression for the contraction parameter $\tilde{\gamma}$ provides sufficient conditions for a fast convergence of EM, which involves an interesting trade off between the SNR $\eta$ and weight $\omega$. More concretely, if the SNR is large enough, the population EM converges fast towards the projection $\overline{\theta}$, which is unique in its absolute value (See Section A.1.4 in the Appendix). This fast convergence of the population EM again enables us to derive the following convergence rate of sample-based EM:

**Corollary 2.** *Consider the model* (11) *such that the assumptions of Theorem 2 hold. For any fixed $\delta \in (0, 1)$, $\theta^0 \in \mathbb{B}(\overline{\theta}, \left|\overline{\theta}\right|/4)$, if $n \geq c_1\log(1/\delta)$ then the sample EM iterates $\theta^{t+1} = M_n(\theta^t)$ satisfy*

$$\left|\theta^t - \overline{\theta}\right| \leq \tilde{\gamma}^t\left|\theta^0 - \overline{\theta}\right| + \frac{c_2}{1-\tilde{\gamma}}\left|\theta^*\right|\left(\theta^{*2} + \sigma^2\right)\sqrt{\frac{\log(1/\delta)}{n}}$$

*with probability at least $1 - \delta$.*

Similar to the structure of the convergence result of sample EM updates in Corollary 1, the result in Corollary 2 also consists of two key terms: the first term is the linear rate of convergence from the population EM operator in Theorem 2 while the second term characterizes the radius of convergence in terms of sample complexity, which is of $\mathcal{O}(\sqrt{1/n})$ after $T = \mathcal{O}(\log n/\log(1/\tilde{\gamma}))$ iterations.

## 4 Robustness of EM for mis-specified variances and weights

In this section, we focus on establishing the convergence rate of EM under different mis-specified regime of the fitted model (1). In particular, we assume that the true data distribution $\mathbb{P}_*$ is given by:

$$\mathbb{P}_* = \frac{1}{2}\mathcal{N}(\theta^*, \sigma^2 - \theta^{*2}) + \frac{1}{2}\mathcal{N}(-\theta^*, \sigma^2 - \theta^{*2}), \tag{14}$$

where $\sigma > 0$ is a given positive number, and $\left|\theta^*\right| \in (0, \sigma/2)$ is a true but unknown parameter. Note that the assumption that $\left|\theta^*\right| \in (0, \sigma/2)$ ensures that the variance $\sigma^2 - \theta^{*2}$ is bounded away from zero. We fit the above model by unbalanced two-component Gaussian location mixture model $\mathbb{P}_\theta$ given by

$$\mathbb{P}_\theta = \pi\mathcal{N}(-\theta, \sigma^2) + (1 - \pi)\mathcal{N}(\theta, \sigma^2), \tag{15}$$

where $\pi := \frac{1}{2}(1 - \epsilon)$ and $\left|\epsilon\right| \in (0, 1)$ are known apriori and only the parameter $\theta$ is to be estimated. In the fitted model $\mathbb{P}_\theta$, we have mis-specified the variance $\sigma^2$ and the weight $\pi$, and we wish to understand the rate of convergence of EM to $\overline{\theta}$, where $\overline{\theta}$ is the parameter of the model $\mathbb{P}_{\overline{\theta}}$, and $\mathbb{P}_{\overline{\theta}}$ is the projection of the true model $\mathbb{P}_*$ onto the model class $\mathcal{P}_\theta := \{\mathbb{P}_\theta : \theta \in \mathbb{R}\}$. We emphasize that the main goal here is to see how the mis-specification with variance and weight affects the statistical inference of EM. We choose variance of the form $\sigma^2 - \theta^{*2}$ because under this setting, we obtain interesting behavior of EM without rendering the proof too technical. We begin with the first result establishing the global linear convergence rate of population EM to $\overline{\theta}$.

**Theorem 3.** *For a two-component Gaussian location mixture model* (14) *and fitted model* (15), *the population EM operator* $\theta \mapsto M(\theta)$ *satisfies*

$$\left| M(\theta) - \bar{\theta} \right| \le \left( 1 - \frac{\epsilon^2}{2} \right) \left| \theta - \bar{\theta} \right|.$$

*Hence, the population EM sequence* $\{\theta^t\}$ *converges geometrically to* $\bar{\theta}$ *from any initialization* $\theta^0$.

There are two interesting features regarding the geometric convergence of population EM updates to $\bar{\theta}$: (1) it does not require an evaluation of bias which was needed for our previous results; (2) it holds under any initialization $\theta^0$. Overall, we have that $\bar{\theta}$ is unique, thereby we conclude that the projection of $\mathbb{P}_*$ to the model class (15) is unique. Before proceeding to the sample-based convergence of EM, we establish the following upper bound on the bias of the parameter $\bar{\theta}$:

**Proposition 3.** *For the two-component Gaussian location mixture model* (14), *we have*

$$\min \left\{ \left| \bar{\theta} - \theta^* \right|, \left| \bar{\theta} + \theta^* \right| \right\} \le c(\theta^*, \sigma) \cdot \sqrt{\left[ 2 \left( 1 - 2\pi \right) \right] \theta^{*\,2} + \theta^{*\,4}},$$

*where* $c(\theta^*, \sigma)$ *is a positive constant depending only on* $\theta^*$, $\sigma$, *and the set* $\Theta$.

Given the above bound, we obtain the range of $\left| \bar{\theta} \right|$ as $\left| \bar{\theta} \right| \in [(1 - C_{\theta^*}) \left| \theta^* \right|, (1 + C_{\theta^*}) \left| \theta^* \right|]$ where $C_{\theta^*} := c(\theta^*, \sigma) \sqrt{\left[ 2(1 - 2\pi) \right] + \theta^{*\,2}}$. Equipped with this bound on $\left| \bar{\theta} \right|$, we have the following result regarding the convergence of sample-based EM:

**Corollary 3.** *Consider the model* (14). *Let radius* $r > 0$ *and* $n \ge c_1 \log(1/\delta)$ *and* $\theta^0 \in \mathbb{B} \left( \bar{\theta}, r \right)$, *then the sample-based EM sequence* $\theta^{t+1} = M_n(\theta^t)$, *satisfies*

$$\left| \theta^t - \bar{\theta} \right| \le \left( 1 - \frac{\epsilon^2}{2} \right)^t \left| \theta^0 - \bar{\theta} \right| + \frac{c_2 \left( (1 + C_{\theta^*}) \left| \theta^* \right| + r \right) \sigma^2}{\epsilon^2} \sqrt{\frac{\log(1/\delta)}{n}},$$

*with probability at least* $1 - \delta$ *where* $\epsilon := 1 - 2\pi$.

The proof of Corollary 3 is similar to those of Corollary 1 or Corollary 2; therefore, it is omitted. The last corollary demonstrates that the sample-based EM iterates converge to ball of radius $\mathcal{O}(\sqrt{1/n})$ around $\bar{\theta}$ after $T = \mathcal{O}(\log n / \log(1/(1 - \epsilon^2/2)))$ iterations.

## 5 Simulation studies

In this section, we illustrate our theoretical results using a few numerical experiments. In particular, we use the EM algorithm to fit 2-component Gaussian mixtures for the three mis-specified settings considered above. For convenience in discussion, we refer to three settings as follows:

- Case 1 refers to the true model (6) from Section 3.1, namely a three component Gaussian mixture where two of the components very close to each other and the quantity $\rho \in (0, 1)$ denotes the extent of weak separation.

- Case 2 refers to the true model (11) from Section 3.2, namely, a three components Gaussian mixture where one of the components has very small weight at origin and the quantity $\omega \in (0, 1)$ denotes the small mixture-weight.

- Finally, Case 3 refers to the true model (14) from Section 4, namely where the true model is a two-Gaussian mixture.

For cases 1 and 2, we fit a symmetric balanced two-Gaussian mixture given by equation (7); while for the third case we fit the unbalanced two-Gaussian mixture given by equation (15) for different values of $\pi$. Let $\widehat{\theta}_n$ denote the final sample EM estimate. Since our results establish that population EM converges to $\bar{\theta}$ (2), we use the final iterate from the population EM sequence to estimate the error $|\widehat{\theta}_n - \bar{\theta}|$. We now summarize our key findings:

   (i) In Figure 1(a), we observe that for all cases the final statistical error $|\widehat{\theta}_n - \bar{\theta}|$ has a parametric rate $n^{-1/2}$ which verifies the claims of Corollaries 1, 2 and 3.

(ii) For all cases, the population EM sequence has a geometric convergence (we omit illustrations for Cases 1 and 2). From Figure 1 (b), we note that for Case 3, the linear convergence of the population EM sequence $\theta^{t+1} = M(\theta^t)$ is affected by the extent of unbalancedness: as $\pi \to 0.5$, the rate of decay of the error of population EM sequence decreases which is consistent with the contraction result stated in Theorem 3.

(iii) In panels (c) and (d) of Figure 1, we plot the biases for Case 1 and 2, with respect to $\rho$ and $\omega$ respectively. Least squares fit on the log-log scale suggest that the biases stated in Proposition 1 and Proposition 2 are potentially sub-optimal: the numerical scaling of the biases $|\theta^* - \overline{\theta}|$ is of the order $\rho^2$ and $\omega$ for Case 1 and 2 respectively, which is significantly smaller than the corresponding scaling of the order $\rho^{1/4}$ and $\omega^{1/8}$ stated in Propositions 1 and 2. In Appendix B, we illustrate the scaling of the bias with $\theta^*$ in these cases via further simulations.

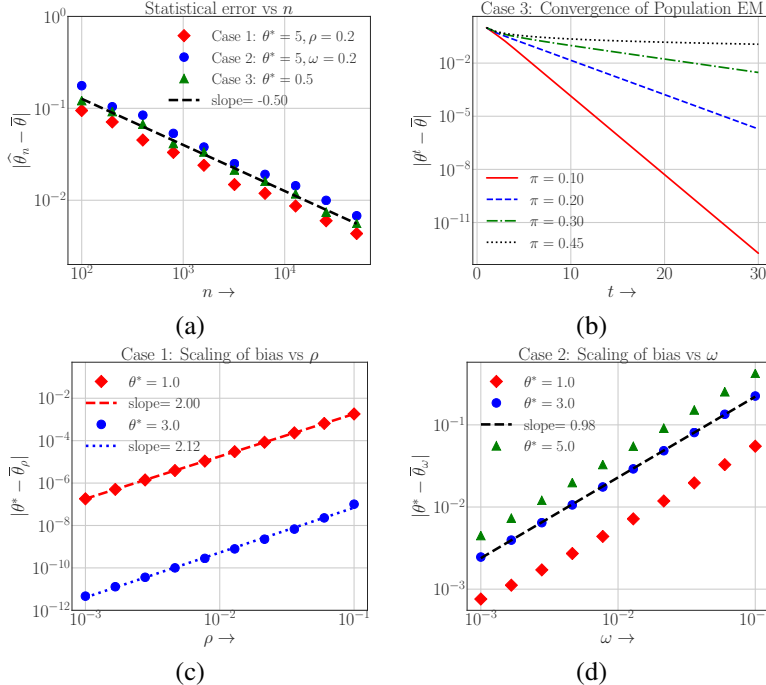

**Figure 1.** Plots depicting behavior of EM when fitting two Gaussian mixture (7) for the three mis-specified mixtures cases (6), (11) and (14), referred to as Case 1, 2 and 3 respectively. (a) For all cases, the statistical error $|\widehat{\theta}_n - \overline{\theta}|$ has the parametric rate $n^{-1/2}$. (b) For Case 3, convergence of population EM sequence $\theta^{t+1} = M(\theta^t)$ is affected by the mixture weight $\pi$. The convergence rate slows down as $\pi \to 0.5$. (c) For Case 1, the bias scales quadratically with the extent of weak-separation $\rho$ for different values of $\theta^*$. (d) For Case 2, the bias scales linearly with the weight $\omega$ of the third component, for different values of $\theta^*$. Refer to the text for more details.

# 6   Discussion

In this paper, we analyzed the behavior of the EM algorithm for certain classes of mis-specified mixture models. Analyzing the behavior of the EM algoirithm under general mis-specification is challenging in general, and we view the results in this paper as a first step towards developing a more general framework for the problem. In this paper, we studied the EM algorithm when it is used to fit Gaussian location mixture models to data generated by mixture models with larger numbers of components, and/or differing mixture weights. We considered only univariate mixtures in this paper, but we believe that several of our results can be extended to multivariate mixtures. It is also interesting to investigate the behavior of the EM algorithm when it is used to fit models with scale parameters that vary (in addition to the location parameters). Besides deriving sharper results for the settings considered in this paper, analyzing the behavior of EM for non-Gaussian and more general mixture models is an appealing avenue for future work.

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
