[Supplementary Material]

# A    Supplementary material

In this appendix, we provide self-contained proofs for our results in the paper. In particular, Section A.1 contains the proofs of our theorems, Section A.2 contains the proof of our propositions and in Section A.3 we prove the corollaries stated in the paper.

## A.1    Proofs for population EM

In this section, we prove our main results on the contraction properties of the population EM algorithm toward the projection onto the model class—namely, Theorems 1, 2 and 3. We treat each of these theorems one-by-one.

### A.1.1    Proof of Theorem 1

The proof of the theorem makes use of Proposition 1 that relates $|\overline{\theta}|$ in terms of $\rho$, $|\theta^*|$, and $\eta = |\theta^*|/\sigma$. Without loss of generality, we assume that $\min\left\{|\theta^* - \overline{\theta}|, |\theta^* + \overline{\theta}|\right\} = |\theta^* - \overline{\theta}|$. For each $u \in [0, 1]$, we define $\theta_u = \overline{\theta} + u(\theta - \overline{\theta})$. Applying Taylor's theorem along the direction $\theta_u$, we obtain that

$$\left|\mathbb{E}\left(2(w_\theta(X) - w_{\overline{\theta}}(X))X\right)\right| = 4\left|\int_0^1 \mathbb{E}\left[\frac{X^2}{\sigma^2(e^{-\theta_u X/\sigma^2} + e^{\theta_u X/\sigma^2})^2}\right](\theta - \overline{\theta})du\right|$$

$$\leq 4\sup_{u \in [0,1]}\{\mathbb{E}[\Gamma_u(X)]\}\left|\theta - \overline{\theta}\right|, \tag{16}$$

where we have defined

$$\Gamma_u(X) := X^2/(\sigma^2(e^{-\theta_u X/\sigma^2} + e^{\theta_u X/\sigma^2})^2), \tag{17}$$

and the expectation is taken over $X$ which is drawn from the Gaussian mixture given by equation (6). Clearly, we have

$$\mathbb{E}[\Gamma_u(X)] = \frac{1}{4}\mathbb{E}_{X \sim \mathcal{N}(-\theta^*(1+\rho),\sigma)}\Gamma_u(X) + \frac{1}{4}\mathbb{E}_{X \sim \mathcal{N}(-\theta^*(1-\rho),\sigma)}\Gamma_u(X) + \frac{1}{2}\mathbb{E}_{X \sim \mathcal{N}(\theta^*,\sigma)}\Gamma_u(X).$$

We now bound the three expectations on the right, which we denote by $T_1, T_2$ and $T_3$ respectively. We claim that

$$T_1, T_2 \leq e^{-\eta^2/64}/16 \quad \text{and} \quad T_3 \leq e^{-\eta^2/64}/8,$$

which in turn implies that $\mathbb{E}[\Gamma_u(X)] \leq \gamma = e^{-\eta^2/64}/4$ and our theorem follows.

We now provide a full derivation for an upper bound on $T_1$. The upper bounds on $T_2$ and $T_3$ can be derived in a similar way and their explicit derivation is omitted here. Letting $R = \text{sign}(\theta_u)$ and $V = -RX/\sigma$, we have

$$D_u := 4T_1 = \mathbb{E}_{X \sim \mathcal{N}(-\theta^*(1+\rho),\sigma)}\Gamma_u(X) = \mathbb{E}[V^2/(e^{-|\theta_u|V/\sigma} + e^{|\theta_u|V/\sigma})^2],$$

where the expectation in the last expression is taken with respect to $V \sim \mathcal{N}(R\theta^*(1+\rho)/\sigma, 1)$. We have

$$D_u \leq \mathbb{E}[V^2 e^{-2|\theta_u|V/\sigma}] \leq \mathbb{E}\left[V^2 e^{-2|\theta_u|V/\sigma}\Big|\mathcal{E}\right] \cdot \mathbb{P}[\mathcal{E}] + \mathbb{E}\left[V^2 e^{-2|\theta_u|V/\sigma}\Big|\mathcal{E}^c\right] \cdot \mathbb{P}[\mathcal{E}^c],$$

where we define the event $\mathcal{E} = \{V|V \leq |\theta^*|(1+\rho)/(4\sigma)\}$. Given a scalar $\mu$, consider the real-valued function $f$ such that $f(t) = t^2 e^{-\mu t}$. Observe that $f(t) \leq \frac{4}{e^2\mu^2}$ for all $t \in \mathbb{R}$ and that $f$ is decreasing on the interval $[2/\mu, \infty)$. Invoking these observations with $\mu = 2|\theta_u|/\sigma$, as long as $|\theta^*|(1+\rho)/(4\sigma) \geq 2/\mu$ or equivalently $|\theta^*|(1+\rho)|\theta_u| \geq 4\sigma^2$, we find that

$$D_u \leq \frac{\sigma^2}{e^2\theta_u^2} \cdot \mathbb{P}[\mathcal{E}] + \frac{\theta^{*2}(1+\rho)^2}{16\sigma^2}e^{-|\theta^*|(1+\rho)|\theta_u|/(2\sigma^2)}. \tag{18}$$

Note that $\theta \in \mathbb{B}(\overline{\theta}, |\overline{\theta}|/4)$ implies that $|\theta_u| \geq 3|\overline{\theta}|/4$ and $\text{sign}(\theta_u) = \text{sign}(\overline{\theta})$ for all $u \in [0, 1]$. Proposition 1 implies that $\overline{\theta} \in [(1 - C_\rho)\theta^*, (1 + C_\rho)\theta^*]$. Since $\rho$ is small enough such that $C_\rho < 1$,

we also have that $\mathrm{sign}(\bar{\theta}) = \mathrm{sign}(\theta^*)$. As a result $\mathbb{E}[V] = \mathrm{sign}(\theta_u)\theta^*(1+\rho)/\sigma = |\theta^*|(1+\rho)/\sigma$. Invoking standard Gaussian tail bounds, we have

$$\mathbb{P}[\mathcal{E}] = \mathbb{P}\left[V - \mathbb{E}[V] \leq -\frac{3}{4}\frac{|\theta^*|(1+\rho)}{\sigma}\right] \leq \exp\left(-\frac{9\theta^{*2}(1+\rho)^2}{32\sigma^2}\right).$$

Plugging this bound along with the fact that $|\theta_u| \geq (1 - C_\rho)|\theta^*|$ in the inequality (18), we find that

$$
\begin{aligned}
D_u &\leq \frac{16\sigma^2 \exp\left(-\frac{9\theta^{*2}(1+\rho)^2}{32\sigma^2}\right)}{9e^2\theta^{*2}(1-C_\rho)^2} + \frac{\theta^{*2}(1+\rho)^2 \exp\left(-\frac{\theta^{*2}(1+\rho)(1-C_\rho)}{8\sigma^2}\right)}{16\sigma^2} \\
&\leq \frac{16}{9e^2}\left(\frac{\sigma^2}{\theta^{*2}(1-C_\rho)^2} + \frac{\theta^{*2}(1+\rho)^2}{\sigma^2}\right)\exp\left(-\frac{\theta^{*2}(1+\rho)(1-C_\rho)}{8\sigma^2}\right) \\
&\leq (\eta^2 + \eta^{-2})\exp(-\eta/16) \quad \text{(since } 64 \leq 9e^2) \\
&\leq 2\eta^2 \exp(-\eta^2/16) \qquad \text{(for } \eta \geq 1) \\
&\leq \exp(-\eta^2/64)/16 \qquad \text{(for } \eta \geq 14),
\end{aligned}
$$

where we have used the fact that $\rho \in (0,1)$ is small enough and that $C_\rho \leq 1/9 < 1/2$. The claim follows.

### A.1.2  Proof of Theorem 2

Equipped with the bounds for the bias term $|\bar{\theta} - \theta^*|$ from Proposition 2, the steps in this proof are similar to the ones used in the proof of Theorem 1. Using Taylor expansion along the direction $\theta_u = \bar{\theta} + u(\theta - \bar{\theta})$ for $u \in [0,1]$, we find that

$$\mathbb{E}[2(w_\theta(X) - w_{\bar{\theta}}(X))X] \leq 4\sup_{u \in [0,1]}\mathbb{E}[\Gamma_u(X)]\,|\theta - \bar{\theta}|, \tag{19}$$

where $\Gamma_u(X)$ is the same term defined above in equation (17). The difference compared to the proof of Theorem 1 is in the distribution of $X$. In particular, now we have

$$
\begin{aligned}
\mathbb{E}[\Gamma_u(X)] &= (1/2 - \omega/2)\mathbb{E}_{X\sim\mathcal{N}(-\theta^*,\sigma)}\Gamma_u(X) + (1/2 - \omega/2)\mathbb{E}_{X\sim\mathcal{N}(\theta^*,\sigma)}\Gamma_u(X) \\
&\quad + \omega\mathbb{E}_{X\sim\mathcal{N}(0,\sigma)}\Gamma_u(X) \\
&= (1/2 - \omega/2)(S_1 + S_2) + \omega S_3.
\end{aligned}
$$

Imitating the steps for bounding $T_1$ in the proof of Theorem 1, we can derive the following bounds for $S_1$ and $S_2$:

$$S_1, S_2 \leq e^{-\eta^2/64}/4,$$

provided that $C(\eta, \omega) := \dfrac{c(\eta^2\sigma^2\omega)^{1/4}}{\sqrt{(1-\omega)}} \leq 1/9 < 1/2$ and $\eta$ is sufficiently large. Thus it is left to provide a bound for the term $S_3$. Using the change of variables $V = \mathrm{sign}(\theta_u)X/\sigma$ and the consequent fact that $V \sim \mathcal{N}(0,1)$ we obtain that

$$S_3 = \mathbb{E}_{X\sim\mathcal{N}(0,\sigma)}[\Gamma(X)] = \mathbb{E}\left[\frac{V^2}{(e^{-|\theta_u|V/\sigma} + e^{|\theta_u|V/\sigma})^2}\right] \overset{(i)}{\leq} \mathbb{E}\left[\frac{V^2}{4}\right] = \frac{1}{4},$$

where step (i) follows from the inequality that $e^{-y} + e^y \geq 2$ for all $y \in \mathbb{R}$. Putting the pieces together yields

$$\mathbb{E}[2(w_\theta(X) - w_{\bar{\theta}}(X))X] \leq (1-\omega)e^{-\eta^2/64} + \omega$$

and we are done.

### A.1.3  Proof of Theorem 3

Using the definition (4a) of the M-update and the self consistency $M(\bar{\theta}) = \bar{\theta}$, we obtain that

$$\left|M(\theta) - M(\bar{\theta})\right| = \underbrace{\left|\mathbb{E}[2(w_\theta(X) - w_{\bar{\theta}}(X))X]\right|}_{=:A}.$$

Note that under the unbalanced mixtures, we have

$$w_\theta(X) = \frac{\pi}{\pi + (1-\pi)e^{-2\theta X/\sigma^2}} \quad \text{and} \quad \frac{\partial}{\partial \theta}(w_\theta(X)) = \frac{2\pi(1-\pi)X/\sigma^2}{(\pi e^{-\theta X/\sigma^2} + (1-\pi)e^{\theta X/\sigma^2})^2}.$$

By means of Taylor expansion along the direction $\theta_u = \bar{\theta} + u(\theta - \bar{\theta})$, the following holds

$$A = 4\pi(1-\pi)\left|\int_0^1 \mathbb{E}\left[\frac{X^2}{\sigma^2\left((1-\pi)\exp\left(-\frac{\theta_u X}{\sigma^2}\right) + \pi\exp\left(\frac{\theta_u X}{\sigma^2}\right)\right)^2}\right] du\right| |\theta - \bar{\theta}|$$

$$\leq 4\pi(1-\pi)|\theta - \bar{\theta}| \max_{u \in [0,1]} \mathbb{E}\left[\Gamma_{\theta_u}(X)\right], \tag{20}$$

where $\Gamma_{\theta_u}(X) := \dfrac{X^2}{\sigma^2\left(\pi\exp\left(-\frac{\theta_u X}{\sigma^2}\right) + (1-\pi)\exp\left(\frac{\theta_u X}{\sigma^2}\right)\right)}$. Let $\pi = \frac{1}{2}(1-\epsilon)$. We claim that

$$\max_{u \in [0,1]} \mathbb{E}\left[\Gamma_{\theta_u}(X)\right] \leq \frac{1 - \epsilon^2/2}{1 - \epsilon^2}, \tag{21}$$

which when plugged in the bound (20) implies that the population EM operator is globally contractive towards $\bar{\theta}$, i.e., $|M(\theta) - M(\bar{\theta})| \leq (1 - \epsilon^2/2)|\theta - \bar{\theta}|$. Therefore, it yields the linear rate of convergence claimed in the theorem.

We now prove the claim (21). Like in proof of Theorem 1, we use $R = \text{sign}(\theta_u)$ and $V = RX/\sigma$. Since $X \sim \frac{1}{2}\mathcal{N}(\theta^*, \sigma^2 - \theta^{*2}) + \frac{1}{2}\mathcal{N}(-\theta^*, \sigma^2 - \theta^{*2})$, it is clear that $\mathbb{E}[V] = 0$ and $\mathbb{E}[V^2] = 1$. By substituting $X = \sigma V/R$, we have

$$\mathbb{E}[\Gamma_{\theta_u}(X)] = \mathbb{E}_V\left[\frac{V^2}{(\pi\exp(-|\theta_u|V/\sigma) + (1-\pi)\exp(|\theta_u|V/\sigma))^2}\right].$$

Now, observe that

$$(\pi e^{-y} + (1-\pi)e^y) \in [\sqrt{(1-\epsilon^2)}, 1], \quad \text{if } e^y \in \left[1, \frac{1+\epsilon}{1-\epsilon}\right], \quad \text{and}$$

$$(\pi e^{-y} + (1-\pi)e^y) > 1, \qquad\qquad \text{otherwise.}$$

Let $\mathcal{E}_{\theta_u}$ denote the event such that $\mathcal{E}_{\theta_u} = \{e^{|\theta_u|V/\sigma} \in [1, (1+\epsilon)/(1-\epsilon)]\}$. Let $\mathcal{E}^c$ and $\mathbb{I}(\mathcal{E})$ respectively denote the complement and the indicator of any event $\mathcal{E}$. Using the observation above and the fact that $\mathbb{E}[V^2] = 1$, we obtain that

$$\mathbb{E}[\Gamma_{\theta_u}(X)] \leq \frac{1}{(1-\epsilon^2)}\mathbb{E}[V^2\,\mathbb{I}(\mathcal{E}_{\theta_u})] + \mathbb{E}[V^2\,\mathbb{I}(\mathcal{E}_{\theta_u}^c)]$$

$$= \frac{1 - \epsilon^2 + \epsilon^2\mathbb{E}[V^2\,\mathbb{I}(\mathcal{E}_{\theta_u})]}{(1-\epsilon^2)}. \tag{22}$$

Note that whenever $\theta_u \neq 0$, we have that

$$\mathbb{E}[V^2\,\mathbb{I}(\mathcal{E}_{\theta_u})] \leq \mathbb{E}[V^2\,\mathbb{I}(V \geq 0)] = \frac{1}{2}. \tag{23}$$

Putting the inequalities (22) and (23) together yields the claim (21).

### A.1.4  Proof of uniqueness of $|\bar{\theta}|$

We provide a proof for the uniqueness of projection in its absolute value from $\mathbb{P}_*$ in (6) or in (11) to the fitted model (7). Due to the similar proof argument between these two cases, we only focus on the case when $\mathbb{P}_*$ is given by (6) and the fitted model is in (7). First, we note that if $\bar{\theta}$ is the projection of $\mathbb{P}_*$ to the fitted model (7), then $-\bar{\theta}$ is also the projection. Therefore, the projection is identifiable in its absolute value. Now, the result of Theorem 1 demonstrates that $\bar{\theta}$ is a unique projection of $\mathbb{P}_*$ to the fitted model (7) within the ball $\mathbb{B}(\bar{\theta}, |\bar{\theta}|/4)$. Based on the inequality (9) and the condition $C_\rho \leq 1/9$, for any two projections $\bar{\theta}_1$ and $\bar{\theta}_2$ of $\mathbb{P}_*$ to the fitted model (7), we find that

$$||\bar{\theta}_1| - |\bar{\theta}_2|| \leq \min\{|\theta^* - \bar{\theta}_1|, |\theta^* + \bar{\theta}_1|\} + \min\{|\theta^* - \bar{\theta}_2|, |\theta^* + \bar{\theta}_2|\} \leq 2C_\rho|\theta^*| \leq 2|\theta^*|/9.$$

Additionally, for any projection $\bar{\theta}$ of $\mathbb{P}_*$ to the fitted model (7), we obtain that $|\bar{\theta}| \in [(1 - C_\rho)|\theta^*|, (1 + C_\rho)|\theta^*|] \in [8|\theta^*|/9, 10|\theta^*|/9]$. Therefore, for any two projections $\bar{\theta}_1$ and $\bar{\theta}_2$ such that $\bar{\theta}_1\bar{\theta}_2 > 0$, we have $\bar{\theta}_2$ in $\mathbb{B}(\bar{\theta}_1, |\bar{\theta}_1|/4)$. On the other hand, for any projections $\bar{\theta}_1$ and $\bar{\theta}_2$ such that $\bar{\theta}_1\bar{\theta}_2 < 0$, we find that $|\bar{\theta}_1 - \bar{\theta}_2| = |\bar{\theta}_1| + |\bar{\theta}_2| > |\bar{\theta}_1|/4$, which proves that $\bar{\theta}_2 \notin \mathbb{B}(\bar{\theta}_1, |\bar{\theta}_1|/4)$. The previous results imply that the projection of $\mathbb{P}_*$ to fitted model (7) is unique in its absolute value.

## A.2 Proofs for computing model biases

In this section, we prove our results on the model bias in different cases, namely Propositions 1, 2 and 3. To facilitate further discussion, we begin with introducing some notations and variational formulation of the Wasserstein distance.

### A.2.1 Notations

Given two distributions $\mathbb{P}$ and $\mathbb{Q}$, we use $h^2(\mathbb{P}, \mathbb{Q})$ and $\mathrm{KL}(\mathbb{P}, \mathbb{Q})$ to denote the hellinger distance and Kullback-Leibler divergence, respectively, between the two distributions. Let $p$ and $q$ denote the corresponding density of these distributions with respect to the Lebesgue measure. Then we have

$$h^2(\mathbb{P}, \mathbb{Q}) = \int (\sqrt{p(x)} - \sqrt{q(x)})^2 dx \quad \text{and} \quad \mathrm{KL}(\mathbb{P}, \mathbb{Q}) = \int p(x) \log \frac{p(x)}{q(x)} dx. \tag{24a}$$

We now introduce some notation to define the Wasserstein distance between two discrete measures. Given any two discrete measures $G = \sum_{i=1}^{k} \pi_i \delta_{\theta_i}$ and $G' = \sum_{i=1}^{k'} \pi_i' \delta_{\theta_i'}$, where $\theta_i, \theta_i' \in \Theta \subset \mathbb{R}$, and $\delta_\theta$ denotes the dirac measure at $\theta$. define the set of couplings $\Pi(G, G')$ between the two measures as follows:

$$\Pi(G, G') = \left\{ T \in \mathbb{R}_+^{k \times k'} : T1_{k'} = \pi, T^\top 1_k = \pi' \right\}, \tag{24b}$$

where $\pi = (\pi_1, \ldots, \pi_k)^T$, $\pi' = (\pi_1', \ldots, \pi_{k'}')^T$, and $1_k$ denotes a $k$-dimensional vector with all entries equal to 1. Put simply, $\Pi(G, G')$ is the set of all joint distributions $T$ on the space $[k] \times [k']$ such that the marginals of the distribution $T$ are equal to $\pi$ and $\pi'$. Furthermore, for any given $r$, define the matrix $D \in \mathbb{R}^{k \times k'}$ of distances between the parameters of $G$ and $G'$ as

$$D_{ij} = \left| \theta_i - \theta_j' \right|^r, \quad (i, j) \in [k] \times [k']. \tag{24c}$$

With these notations in place, the Wasserstein distance [14] of order $r \geq 1$ between the two measures $G$ and $G'$ is given by

$$W_r^r(G, G') := \inf_{T \in \Pi(G, G')} \sum_{i=1}^{k} \sum_{j=1}^{k'} T_{ij} D_{ij}. \tag{24d}$$

With this notation in place, we now turn to the proofs of our propositions.

### A.2.2 Proof of Proposition 1

In order to prove this proposition, we utilize several bounds between KL divergence, Hellinger distance, and Wasserstein distance. The road-map of the proof is as follows: First, we relate the KL divergences between the mixture distributions to the Wasserstein distances between the corresponding discrete mixing measures. Then, using carefully constructed couplings, we derive lower and upper bounds on the Wasserstein distances in terms of the bias term $|\bar{\theta} - \theta^*|$ and other problem parameters to obtain the claimed result.

For any mixing-measure (discrete mixture measure) $G$ on $\Theta$, let $\mathbb{P}_G$ denote the Gaussian mixture distribution induced by $G$ on $\mathbb{R}$ whose density is given by $p_G(x) = \int_\Theta \phi(x; \theta, \sigma) dG$, where $\phi(\cdot; \theta, \sigma)$ denotes the density of the Gaussian distribution $\mathcal{N}(\theta, \sigma^2)$. We introduce the following notation for the mixing-measures:

$$G_* = \frac{1}{4}\delta_{-\theta^*(1-\rho)} + \frac{1}{4}\delta_{-\theta^*(1+\rho)} + \frac{1}{2}\delta_{\theta^*}, \quad \text{and} \quad G(\theta) = \frac{1}{2}\delta_{-\theta} + \frac{1}{2}\delta_\theta. \tag{25a}$$

Note that in our notation, $\mathbb{P}_* = \mathbb{P}_{G_*}$, $G_* \neq G(\theta^*)$ and consequently $\mathbb{P}_{G_*} \neq \mathbb{P}_{G(\theta^*)}$. Define

$$\bar{G} := G(\bar{\theta}) \quad \text{where} \quad \bar{\theta} \in \arg\min_{\theta \in \Theta} \mathrm{KL}(\mathbb{P}_{G_*}, \mathbb{P}_{G(\theta)}). \tag{25b}$$

Applying Lemma 1 from the paper [10], we obtain the following relationship between the KL divergence between the Gaussian-mixture measures $\mathbb{P}_{G^*}$ and $\mathbb{P}_{G(\theta)}$ and the Wasserstein distance between the corresponding mixing measures $G$ and $G(\theta)$:

$$\mathrm{KL}(\mathbb{P}_{G_*}, p_{G(\theta)}) \leq W_2^2(G_*, G(\theta))/(2\sigma^2) \quad \text{for any } \theta \in \Theta.$$

Consequently, we find that

$$\mathrm{KL}(\mathbb{P}_{G_*}, \mathbb{P}_{\bar{G}}) = \min_{\theta \in \Theta} \mathrm{KL}(\mathbb{P}_{G_*}, \mathbb{P}_{G(\theta)}) \leq \min_{\theta \in \Theta} W_2^2(G_*, G(\theta))/(2\sigma^2). \tag{26}$$

On the other hand, from the classical bound between KL divergence and Hellinger distance, we have

$$\mathrm{KL}(\mathbb{P}_{G_*}, \mathbb{P}_{\bar{G}}) \geq 2h^2(\mathbb{P}_{G_*}, \mathbb{P}_{\bar{G}}). \tag{27}$$

Noting that, the univariate location Gaussian distribution is 4-strongly identifiable (cf. Definition 2.2 in [6] for the definition of 4-strongly identifiable condition and Theorem 2.4 in [6] for the result with univariate location Gaussian), with an application of the result of Theorem 6.3 in [6], we obtain that

$$h^2(\mathbb{P}_{G_*}, \mathbb{P}_{\bar{G}}) \geq CW_2^8(G_*, \bar{G}), \tag{28}$$

where $C$ is a universal constant depending only on $\Theta$. The results from (26), (27), and (28) lead to

$$2CW_2^8(G_*, \bar{G}) \leq \mathrm{KL}(\mathbb{P}_{G_*}, \mathbb{P}_{\bar{G}}) = \min_{\theta \in \Theta} \mathrm{KL}(\mathbb{P}_{G_*}, \mathbb{P}_{G(\theta)}) \leq \frac{1}{2\sigma^2} \min_{\theta \in \Theta} W_2^2(G_*, G(\theta)),$$

which implies that

$$2\sigma\sqrt{C}W_2^4(G_*, \bar{G}) \leq \min_{\theta \in \Theta} W_2(G_*, G(\theta)) \leq W_2(G_*, G(\theta^*)). \tag{29}$$

(Recall in our notation $G(\theta^*) \neq G_*$.) Now we derive obtain an upper bound for the distance $W_2(G_*, \bar{G})$, by deriving an upper bound for the distance $W_2(G_*, G(\theta^*))$ using the variational formulation (24d) of the Wasserstein distance. In particular, we use a particular coupling to derive an upper bound for $W_2^2(G_*, G(\theta^*))$. Recalling the definitions (24b) and (24c) of the coupling $\Pi(G_*, G(\theta^*))$ and the corresponding distance matrix $D$ for $G = G_*, G' = G(\theta^*), r = 2$, we find that

$$T = \begin{bmatrix} 1/4 & 0 \\ 1/4 & 0 \\ 0 & 1/2 \end{bmatrix} \in \Pi(G_*, G(\theta^*)) \quad \text{and} \quad D = \begin{bmatrix} \rho^2\theta^{*2} & (2-\rho)^2\theta^{*2} \\ \rho^2\theta^{*2} & (2+\rho)^2\theta^{*2} \\ 4\theta^{*2} & 0 \end{bmatrix}.$$

Now applying the definition (24d), we obtain that

$$W_2^2(G_*, G(\theta^*)) \leq \sum_{i=1}^{3}\sum_{j=1}^{2} T_{ij}D_{ij} = \frac{1}{4}\rho^2\theta^{*2} + \frac{1}{4}\rho^2\theta^{*2} + \frac{1}{2}0 = \rho^2\theta^{*2}. \tag{30}$$

Putting the previous inequalities (29) and (30) together, we conclude that

$$W_2(G_*, \bar{G}) \leq c \left( \frac{\rho |\theta^*|}{\sigma} \right)^{1/4}, \tag{31}$$

where $c = 1/(4C)^{1/8}$ is a universal positive constant that depends only on the set $\Theta$.

Now we directly obtain a lower bound for $W_2(G_*, \bar{G})$ by invoking the definition (24d) for the pair $(G_*, \bar{G})$. The corresponding distance matrix is given by

$$\bar{D} = \begin{bmatrix} \left(-\bar{\theta} + \theta^*(1-\rho)\right)^2 & \left(\bar{\theta} + \theta^*(1-\rho)\right)^2 \\ \left(-\bar{\theta} + \theta^*(1+\rho)\right)^2 & \left(\bar{\theta} + \theta^*(1+\rho)\right)^2 \\ \left(-\bar{\theta} + \theta^*\right)^2 & \left(\bar{\theta} + \theta^*\right)^2. \end{bmatrix}.$$

Noting that for any coupling $\bar{T} \in \Pi(G_*, \bar{G})$, we have $\bar{T}_{ij} \geq 0, \sum_{i,j} \bar{T}_{ij} = 1$, we have that

$$\sum_{i,j} \bar{T}_{ij} \bar{D}_{ij} \geq \min_{i,,j} \bar{D}_{ij},$$

and hence

$$\begin{aligned} W_2(G_*, \bar{G}) \geq \min_{i,,j} \sqrt{\bar{D}_{ij}} &= \min \left\{ \left| -\bar{\theta} + \theta^*(1+\rho) \right|, \left| -\bar{\theta} + \theta^*(1-\rho) \right|, \left| -\bar{\theta} + \theta^* \right| \right. \\ &\qquad \left. \left| \bar{\theta} + \theta^*(1+\rho) \right|, \left| \bar{\theta} + \theta^*(1-\rho) \right|, \left| \bar{\theta} + \theta^* \right| \right\} \\ &\geq \min \left\{ \left| \theta^* - \bar{\theta} \right|, \left| \theta^* + \bar{\theta} \right| \right\} - \rho \left| \theta^* \right|, \end{aligned} \tag{32}$$

where the last step follows from the triangle inequality. Putting the inequalities (31) and (32) together yields that

$$\min \left\{ \left| \theta^* - \bar{\theta} \right|, \left| \theta^* + \bar{\theta} \right| \right\} \leq \rho \left| \theta^* \right| + c \left( \frac{\rho \left| \theta^* \right|}{\sigma} \right)^{1/4},$$

and the proposition follows.

### A.2.3 Proof of Proposition 2

The proof of the proposition is similar to the proof of Proposition 1 except for a few differences which we point to now. For any mixing-measure (discrete mixture measure) $G$ on $\Theta$, we denote by $\mathbb{P}_G$ the corresponding Gaussian mixture distribution with density $p_G(x) = \int_\Theta \phi(x; \theta, \sigma) dG$, where $\phi(\cdot; \theta, \sigma)$ denotes the density of the Gaussian distribution $\mathcal{N}(\theta, \sigma^2)$. For this proof, we define

$$G_* = \frac{(1-\omega)}{2} \delta_{-\theta^*} + \omega \delta_0 + \frac{(1-\omega)}{2} \delta_{\theta^*} \quad \text{and} \quad G(\theta) = \frac{1}{2} \delta_{(-\theta, \sigma)} + \frac{1}{2} \delta_{(\theta, \sigma)}.$$

Once again in our notation, we have $\mathbb{P}_* = \mathbb{P}_{G_*}$, $G_* \neq G(\theta^*)$ and consequently $\mathbb{P}_{G_*} \neq \mathbb{P}_{G(\theta^*)}$. Rewriting equation (29), we have

$$2\sigma \sqrt{C} W_2^4(G_*, \bar{G}) \leq \min_{\theta \in \Theta} W_2(G_*, G(\theta)) \leq W_2(G_*, G(\theta^*)),$$

where $C$ is some universal positive constant only depending on $\Theta$. Once again, we derive an upper bound for $W_2(G_*, \bar{G})$ by deriving an upper bound on $W_2(G_*, G(\theta^*))$. We now provide a coupling $T$ and the matrix $D$ (refer to equations (24b),(24c)) for the pair $G_*, G(\theta^*)$:

$$T = \begin{bmatrix} (1-\omega)/2 & 0 \\ \omega/2 & \omega/2 \\ 0 & (1-\omega)/2 \end{bmatrix} \in \Pi(G_*, G(\theta^*)) \quad \text{and} \quad D = \begin{bmatrix} 0 & 4\theta^{*2} \\ \theta^{*2} & \theta^{*2} \\ 4\theta^{*2} & 0 \end{bmatrix}.$$

Using the definition (24d), we have that $\sum_{ij} T_{ij} D_{ij}$ is an upper bound for $W_2^2(G_*, G(\theta^*))$. Doing some algebra yields that

$$W_2(G_*, \bar{G}) \leq c W_2^{1/4}(G_*, G(\theta^*)) \leq c \frac{\omega^{1/8} \left| \theta^* \right|^{1/4}}{\sigma^{1/4}} \tag{33}$$

where $c = 1/(4C)^{1/8}$ is a universal positive constant that only depends on $\Theta$.

Now for the lower bound on $W_2(G_*, \bar{G})$, suppose that we are a given coupling $\bar{T} \in \Pi(G_*, \bar{G})$, and the distance matrix with elements

$$\bar{D} = \begin{bmatrix} \left(\theta^* - \bar{\theta}\right)^2 & \left(\theta^* + \bar{\theta}\right)^2 \\ \bar{\theta}^2 & \bar{\theta}^2 \\ \left(\theta^* + \bar{\theta}\right)^2 & \left(\theta^* - \bar{\theta}\right)^2 \end{bmatrix}.$$

Direct computation leads to

$$\begin{aligned} \sum_{ij} \bar{T}_{ij} \bar{D}_{ij} &= (\bar{T}_{11} + \bar{T}_{32}) \left(\theta^* - \bar{\theta}\right)^2 + (\bar{T}_{12} + \bar{T}_{31}) \left(\theta^* + \bar{\theta}\right)^2 + (T_{21} + T_{22}) \bar{\theta}^2 \\ &\geq (\bar{T}_{11} + \bar{T}_{32} + \bar{T}_{12} + \bar{T}_{31}) \cdot \min \left\{ \left(\bar{\theta} + \theta^*\right)^2, \left(\bar{\theta} - \theta^*\right)^2 \right\} \\ &= (1-\omega) \min \left\{ \left(\bar{\theta} + \theta^*\right)^2, \left(\bar{\theta} - \theta^*\right)^2 \right\}, \end{aligned}$$

where the final inequality is due to the constraint $\bar{T}_{11} + T_{21} = T_{13} + T_{23} = (1 - \omega)/2$. As a result, we have

$$W_2^2(G_*, \bar{G}) \geq (1 - \omega) \min \left\{ \left( \bar{\theta} + \theta^* \right)^2, \left( \bar{\theta} - \theta^* \right)^2 \right\}. \tag{34}$$

Combining the inequalities (33) and (34), we obtain that

$$\min \left\{ \left| \bar{\theta} - \theta^* \right|, \left| \bar{\theta} + \theta^* \right| \right\} \leq \frac{c\omega^{1/8} \left| \theta^* \right|^{1/4}}{\sigma^{1/4} \sqrt{1 - \omega}},$$

thereby yielding the desired result.

### A.2.4  Proof of Proposition 3

While the proof of the proposition follows several ideas from the other proofs on controlling the biases, a key difference for this case is that the two measures do not have same variance, and that forces us to use a couple new ideas in the proof. Define

$$G_* = \frac{1}{2}\delta_{(-\theta^*, \sigma^2 - \theta^{*2})} + \frac{1}{2}\delta_{(\theta^*, \sigma^2 - \theta^{*2})} \quad \text{and} \quad G(\theta) = \pi\delta_{(-\theta, \sigma^2)} + (1 - \pi)\frac{1}{2}\delta_{(\theta, \sigma^2)}.$$

Note that we have $G_* \neq G(\theta^*)$. Unlike the cases considered in Proposition 1 and 2, the key lower bound $h^2(\mathbb{P}_{G_*}, \mathbb{P}_{\bar{G}}) \geq CW_2^8(G_*, \bar{G})$ does not apply hear for $C$ some universal constant depending only on $\Theta$. Such an issue is caused due to the fact that the variance of the components corresponding to $G_*$ and $\bar{G}$ are different as $\theta^* \neq 0$. To overcome this issue, we claim the following point-wise bound:

$$h(\mathbb{P}_{G_*}, \mathbb{P}_{\bar{G}}) \geq C(G_*)W_2(G_*, \bar{G}), \tag{35}$$

where $C(G_*)$ is a positive constant depending only on $G_*$ and $\Theta$. To simplify notation, we substitute $C = C(G_*)$. Deferring the proof of the claim (35) to the end of this section, we proceed to finishing the proof.

Note that the relationship (27) between the Hellinger distance and the KL divergence is still valid but the bound (26) needs to be modified as follows (again applying Lemma 1 from the paper [10]):

$$\mathrm{KL}(\mathbb{P}_{G_*}, \mathbb{P}_{\bar{G}}) = \min_{\theta \in \Theta} \mathrm{KL}(\mathbb{P}_{G_*}, \mathbb{P}_{G(\theta)}) \leq \min_{\theta \in \Theta} \frac{C}{\sigma^2} W_2^2(G_*, G(\theta)) \tag{36}$$

for some large constant $C$. Putting the pieces together yields that

$$W_2(G_*, \bar{G}) \leq \frac{C}{\sigma} W_2(G_*, G(\theta^*)).$$

We now consider the following coupling and the distance matrix (refer to equations (24b) and (24c) respectively) for the mixing-measure pair $(G_*, G(\theta^*))$:

$$T = \begin{bmatrix} \pi & 0 \\ (1/2 - \pi) & 1/2 \end{bmatrix} \in \Pi(G_*, G(\theta^*)) \quad \text{and} \quad D = \begin{bmatrix} \theta^{*4} & 4\theta^{*2} + \theta^{*4} \\ 4\theta^{*2} + \theta^{*4} & \theta^{*4}. \end{bmatrix}$$

Invoking the variational formulation (24d), we find that

$$W_2^2(G_*, G(\theta^*)) \leq \sum_{i,j} T_{ij} D_{ij} = (2 - 4\pi)\theta^{*2} + \theta^{*4}.$$

Therefore, the following inequality holds

$$W_2(G_*, \bar{G}) \leq \frac{c(\theta^*)}{\sigma} \sqrt{(2 - 4\pi)\theta^{*2} + \theta^{*4}} \tag{37}$$

where $c(\theta^*)$ is a positive constant that only depends on $G_*$ and $\Theta$. On the other hand, for any coupling $\bar{T} \in \Pi(G_*, \bar{G})$, arguing as in the previous proofs, we have that

$$\sum_{i,j} \bar{T}_{ij} \bar{D}_{ij} \geq \min_{i,j} \bar{D}_{ij} \quad \text{where} \quad \bar{D} = \begin{bmatrix} \left(\theta^* - \bar{\theta}\right)^2 + \theta^{*4} & \left(\theta^* + \bar{\theta}\right)^2 + \theta^{*4} \\ \left(\theta^* + \bar{\theta}\right)^2 + \theta^{*4} & \left(\theta^* - \bar{\theta}\right)^2 + \theta^{*4} \end{bmatrix}$$

and consequently we have that

$$W_2(G_*, \bar{G}) \geq \min_{i,j} \sqrt{\bar{D}_{ij}} \geq \min \left\{ \left| \bar{\theta} - \theta^* \right|, \left| \bar{\theta} + \theta^* \right| \right\}. \tag{38}$$

Combining the inequalities (37) and (38) yields the result.

**Proof of claim** (35)   In order to prove inequality (35), it suffices to show that

$$\inf_{\theta \in \Theta} h(\mathbb{P}_{G_*}, \mathbb{P}_{G(\theta)})/W_2(G_*, G(\theta)) > 0.$$

We proceed via proof by contraction: assume that the above bound does not hold. It implies that we can find a sequence of $\{\theta_n\}_{n \geq 1}$ such that $h(\mathbb{P}_{G_*}, \mathbb{P}_{G(\theta_n)})/W_2(G_*, G(\theta_n)) \to 0$ as $n \to \infty$. Since $\Theta$ is a compact subset of $\mathbb{R}$, there must exist a subsequence of $\theta_n$ such that $\theta_n \to \theta'$ for some $\theta' \in \Theta$. Without loss of generality, we can replace this subsequence of $\theta_n$ by its whole sequence. Applying Fatou's lemma, we obtain that

$$0 = \lim_{n \to \infty} h^2(\mathbb{P}_{G_*}, \mathbb{P}_{G(\theta_n)}) = \frac{1}{2} \int \liminf_{n \to \infty} \left( \sqrt{\mathbb{P}_{G_*}(x)} - \sqrt{\mathbb{P}_{G(\theta_n)}(x)} \right)^2 d\mu(x)$$
$$= h^2(\mathbb{P}_{G_*}, \mathbb{P}_{G(\theta')}).$$

The above result implies that $\mathbb{P}_{G_*}(x) = \mathbb{P}_{G(\theta')}(x)$ almost surely. Due to the general identifiability of finite location-scale Gaussian mixtures [11], the previous equations implies that $G_* \equiv G(\theta')$, which is a contradiction as $\pi \in (0, 1/2)$ and $\theta^* \neq 0$. Therefore, we have established the claim (35).

## A.3   Proofs for sample-based EM

In this section, we prove the Corollaries 1 and 2. The proof for Corollary 3 is rather similar and is omitted.

### A.3.1   Proof of Corollary 1

To prove this corollary, we use Theorem 2 by Balakrishnan et al. [1] and note that it suffices to establish the following lemma:

**Lemma 1.** *For any threshold $\delta \in (0, 1)$, we have*

$$\mathbb{P} \left[ \sup_{\theta \in \Omega} |M_n(\theta) - M(\theta)| - c_2 (1 + C_\rho) |\theta^*| \left( (1 + \rho^2)\theta^{*2} + \sigma^2 \right) \sqrt{\frac{\log(1/\delta)}{n}} \geq 0 \right] \leq \delta,$$

*where $\Omega := \mathbb{B}(\overline{\theta}, |\overline{\theta}|/4)$ for sample size $n \geq c_1 \log(1/\delta)$ where $c_1$ and $c_2$ are universal positive constants.*

*Proof.* The proof of this lemma makes use of standard arguments to derive Rademacher complexity bounds. We denote

$$Z = \sup_{\theta \in \Omega} |M_n(\theta) - M(\theta)|.$$

By means of standard symmetrization argument with empirical processes [12], the following holds

$$\mathbb{E}\left[ \exp\left( \lambda Z \right) \right] \leq \mathbb{E} \left[ \exp \left( \sup_{\theta \in \Omega} \frac{2\lambda}{n} \left| \sum_{i=1}^{n} \epsilon_i (2w_\theta(X_i) - 1) X_i \right| \right) \right], \text{ for any } \lambda > 0.$$

For any $\theta$ and $\theta'$, we have

$$|2w_\theta(x) - 2w'_\theta(x)| \leq |\theta - \theta'| |x|,$$

for all $x \in \mathbb{R}$. Invoking the Ledoux-Talagrand contraction result for Rademacher processes [8] yields

$$\mathbb{E} \left[ \exp \left( \sup_{\theta \in \Omega} \frac{2\lambda}{n} \left| \sum_{i=1}^{n} \epsilon_i (2w_\theta(X_i) - 1) X_i \right| \right) \right] \leq \mathbb{E} \left[ \exp \left( \sup_{\theta \in \Omega} \frac{4\lambda}{n} \left| \sum_{i=1}^{n} \epsilon_i X_i^2 \theta \right| \right) \right]$$
$$= \mathbb{E} \left[ \exp \left( \frac{5\lambda |\overline{\theta}|}{n} \sum_{i=1}^{n} \epsilon_i X_i^2 \right) \right], \qquad (39)$$

where $\epsilon_1, \ldots, \epsilon_n$ are i.i.d. Rademacher random variables which are independent of $X_i$'s. Recalling the distribution of $X_i$'s, we have

$$
\begin{aligned}
\mathbb{E}\left[\exp\left(\lambda X_i\right)\right] &= \exp\left(\lambda^2 \sigma^2/2\right)\left(\frac{1}{4}\exp\left(-\lambda\theta^*(1+\rho)\right) + \frac{1}{4}\exp\left(-\lambda\theta^*(1-\rho)\right) + \frac{1}{2}\exp(\lambda\theta^*)\right) \\
&\overset{(i)}{\leq} \exp\left(\lambda^2 \sigma^2/2\right)\left(\frac{1}{4}\exp\left(-\lambda\theta^*\rho\right) + \frac{1}{4}\exp\left(\lambda\theta^*\rho\right)\right)\left(\exp(-\lambda\theta^*) + \exp(\lambda\theta^*)\right) \\
&\overset{(ii)}{\leq} \exp\left(\lambda^2 \frac{(1+\rho^2)\theta^{*2} + \sigma^2}{2}\right),
\end{aligned}
$$

where step (i) and step (ii), respectively, follow from the basic inequalities $\exp(-y) + \exp(y) \geq 2$ and $\exp(-y) + \exp(y) \leq 2\exp(y^2/2)$ for all $y \in \mathbb{R}$. Thus, the random variable $X_i$ is sub-Gaussian with parameter at most $\gamma = ((1+\rho^2)\theta^{*2} + \sigma^2)^{1/2}$ for any $i \in [n]$. Since any squared sub-Gaussian random variable is a sub-exponential random variable, the following inequality holds [13]:

$$
\mathbb{E}\left[\exp\left(tX_i^2 - t\mathbb{E}\left[X_i^2\right]\right)\right] \leq \exp\left[16t^2\gamma^4\right] \text{ for all } |t| \leq \frac{1}{4\gamma^2}.
$$

Furthermore, we can bound the second moment of $X_i$ as follows:

$$
\begin{aligned}
\mathbb{E}\left[X_i^2\right] &= \frac{1}{4}\left((\theta^*(1+\rho))^2 + \sigma^2\right) + \frac{1}{4}\left((\theta^*(1-\rho))^2 + \sigma^2\right) + \frac{1}{2}\left(\theta^{*2} + \sigma^2\right) \\
&\leq (1+\rho^2)\theta^{*2} + \sigma^2 = \gamma^2.
\end{aligned}
$$

Using these MGF and moment bounds, we find that

$$
\begin{aligned}
\mathbb{E}\left[\exp\left(t\epsilon_i X_i^2\right)\right] &= \frac{1}{2}\mathbb{E}\left[\exp\left(tX_i^2\right)\right] + \frac{1}{2}\mathbb{E}\left[\exp\left(-tX_i^2\right)\right] \\
&\leq \exp\left(16t^2\gamma^4\right)\frac{1}{2}\left(\exp(t\gamma^2) + \exp(-t\gamma^2)\right) \\
&\leq \exp(17t^2\gamma^4),
\end{aligned}
\tag{40}
$$

for all $|t| \leq \frac{1}{4\gamma^2}$. Plugging in $t = 5\lambda\left|\overline{\theta}\right|/n$ in the bound (40) and combining with the bound (39) yields the following MGF bound

$$
\mathbb{E}\left[\exp\left(\lambda Z\right)\right] \leq \exp\left(425\lambda^2\overline{\theta}^2\gamma^4/n\right) \leq \exp\left(425\lambda^2\left(1+C_\rho\right)^2\theta^{*2}\gamma^4/n\right)
$$

for $|\lambda| \leq n/(20\gamma^2\left|\overline{\theta}\right|)$. Here the second inequality in the above display is due to the upper bound $\left|\overline{\theta}\right| \leq (1+C_\rho)\left|\theta^*\right|$ from Proposition 1. By virtue of standard Chernoff's approach, the above MGF bound implies that

$$
Z \leq c_2\left(1+C_\rho\right)\left|\theta^*\right|\gamma^2\sqrt{\frac{\log(1/\delta)}{n}} = c_2\left(1+C_\rho\right)\left|\theta^*\right|\left((1+\rho^2)\theta^{*2} + \sigma^2\right)\sqrt{\frac{\log(1/\delta)}{n}},
$$

with probability at least $1-\delta$ as long as $n \geq c_1\log(1/\delta)$ for sufficiently large positive constants $c_1$ and $c_2$. The lemma now follows. $\qquad \square$

### A.3.2 Proof of Corollary 2

Similar to the argument of Corollary 1, to prove this corollary it is sufficient to establish the following lemma:

**Lemma 2.** *For any threshold $\delta \in (0,1)$, we have*

$$
\mathbb{P}\left[\sup_{\theta \in \Omega}\left|M_n(\theta) - M(\theta)\right| - c_2\left(1+C_\omega\right)\left|\theta^*\right|\left(\theta^{*2} + \sigma^2\right)\sqrt{\frac{\log(1/\delta)}{n}} \geq 0\right] \leq \delta,
\tag{41}
$$

*where $\Omega := \mathbb{B}(\overline{\theta}, \left|\overline{\theta}\right|/4)$ for sample size $n \geq c_1\log(1/\delta)$ where $c_1$ and $c_2$ are universal positive constants.*

*Proof.* Following the argument used in the proof of Lemma 1, we obtain that

$$\mathbb{E}\left[\exp\left(\lambda Z\right)\right] \leq \mathbb{E}\left[\exp\left(\frac{5\lambda\left|\overline{\theta}\right|}{n}\sum_{i=1}^{n}\epsilon_i X_i^2\right)\right],$$

where $Z = \sup_{\theta \in \Omega}\left|M_n(\theta) - M(\theta)\right|$ and $\epsilon_1, \ldots, \epsilon_n$ are i.i.d. Rademacher random variables independent of $X_i$'s. Recalling the distribution of $X_i$'s, we have

$$\mathbb{E}\left[\exp\left(\lambda X_i\right)\right] = \exp\left(\lambda\sigma^2/2\right)\left(\left(\frac{1-\omega}{2}\right)\left(\exp\left(-\lambda\theta^*\right) + \exp\left(\lambda\theta^*\right)\right) + \omega\right)$$

$$\leq \exp\left(\lambda^2\sigma^2/2\right)\left((1-\omega)\exp\left(\frac{\lambda^2\theta^{*2}}{2}\right) + \omega\right)$$

$$\leq \exp\left(\lambda^2(\sigma^2 + \theta^{*2})/2\right).$$

Thus, the random variables $X_i$ are independent sub-Gaussian with parameter at most $\gamma = \sqrt{\theta^{*2} + \sigma^2}$ for all $i \in [n]$. Furthermore, we can bound the second moment of $X_i$ as follows:

$$\mathbb{E}\left[X_i^2\right] = (1-\omega)(\theta^{*2} + \sigma^2) + \omega\sigma^2 \leq (\theta^{*2} + \sigma^2). \tag{42a}$$

Using these MGF and moment bounds, we have

$$\mathbb{E}\left[\exp\left(t\epsilon_i X_i^2\right)\right] \leq \exp(17t^2\gamma^4) \quad \text{for all } |t| \leq 1/4\gamma^2. \tag{42b}$$

Finally, performing computations similar to those in the proof of Lemma 1 yields the claim.

$\square$

# B  Further numerical experiments

Here we provide supplementary material for the numerical experiments presented in Section 5 of the main text. In particular, we numerically illustrate the scalings of the bias $\left|\theta^* - \overline{\theta}\right|$ as a function of $\theta^*$ in Figure 2. We use population EM (the final iterate) to estimate $\overline{\theta}$ (2) for the different settings. We simulated two different settings for Case 1 corresponding to the two mixture fit (7) for the three mixture model (6) and Case 2 corresponding to the two mixture fit (7) for the three mixture model (11) and report the results in Figure 2. The behavior of the bias $\left|\theta^* - \overline{\theta}\right|$ is rather different in the two cases. We see that for Case 1 the bias decreases with increase in $\theta^*$, while for Case 2, it increases with increase in $\theta^*$. Such a behavior is not captured in our results stated in Propositions 1 and 2. Thus, the bias analysis presented in this paper should be considered only a first step towards understanding the under-fitted mixtures. Providing a sharper framework that yields optimal bounds for such biases remains an interesting future direction.

**Figure 2.** Plots of the best two mixture Gaussian fits for the data generated from three Gaussian mixtures. In (a) & (b), we consider two settings of Case 1 (6) and in panels (c) & (d), two settings of Case 2 (11). We see that for Case 1 as $\theta^*$ increases, $\overline{\theta} \to \theta^*$ and for Case 2 an increase in $\theta^*$ leads to an increase in the bias $|\theta^* - \overline{\theta}|$. Indeed as we plot the bias term in panels (e) and (f), we see that for Case 1, larger $\theta^*$ has a smaller bias and on the contrary for Case 2, the bias increases with increase in $\theta^*$.