[Reviews · NeurIPS 2018]

Reviewer 1



The authors focus on the behavior of EM when model is misspecified, obtaining insights form a number of restricted cases. This is an important topic, first because EM is a very popular tool, and second because misspecification must be the rule rather than the exception. The author do have some interesting derivations and results, and they do contribution with some novel insights. However, there are some questions here concerning significance. The main criticism here is that the studied cases are really special ones; mostly one is looking at mixtures of two Gaussians with known variance and expectations theta and -theta (and then corrupted versions of this model). Of course one must study the easy cases first, but here the cases are so specific that I wonder what can be extrapolated from them. This is the main difficulty. Also, I find that the meaning of "misspecified" is not always clear. Misspecification means that a class of models is selected, say belonging to set Theta containing values for a parameter theta, and the true parameter is not in Theta. This seems to be used in the paper, but sometimes the authors write as if an estimated value is misspecified just because it is not equal to the true value. Is it really a case of misspecification? I guess not. But perhaps the authors do not think that either; perhaps it is just the text that seems to pose things this way. The problem seems to be that the authors do not clarify what exactly is supposed to be known and what is estimated, hence the reader sees "variance is misspecified" and thinks "well, but variance is estimated, how can it be misspecified" only to find that variance is supposed known (a rather strong assumption!!). In any case, most of the time I was comfortable with the definition of misspecification I just gave; my suggestion is that the definition of misspecification should be formally given, together with more details on what is known and what is estimated. Now the thing is that results concerning the behavior of maximum likelihood under misspecification are rather old; for example there is classic work by Berk from 1966. The idea there is that by looking for maximum likelihood one finds the closest projection via Kullback-Leibler divergence; this is a *consequence* of maximum likelihood. But in this paper the closest projection appears suddenly, as if it were an obvious goal in itself. But the reader must wonder why one is looking at the projection via Kullback-Leibler; more explanation should be provided. Question: why is the condition \eta>=1 necessary? What does it guarantee? Also, the constant \gamma should be defined and explained in the statement of Theorem 1, not after it. I had trouble understanding the meaning of Figures 1,2,3: what exactly is depicted there? What do they show? Please explain better. Concerning the text, a few points: - Abstract, line -3: "implies". - Order references when a set of them appears together. - Section 3: first paragraph is not clear; sentence "Consequently, the..." is really hard to parse. - There are problems with capitalization in the references (em, gaussian).

Reviewer 2



# After authors response I have read the authors’ responses about the references and some intuitions about the results and the model and the model in section 4, and I thank the authors for their explanations. I made some detailed comments about the figures and their interpretation, that go beyond a simple size problem. I acknowledge the will of the authors to revisit both their display and exploitation. I also acknowledge the authors’ intention to work on the exposition and redaction in some places, previously signaled. I thank the authors for receiving these comments, I think that the paper would highly benefit from an improvement on theses matters. # Summary In this paper, the authors are interested in the effect of model specifications on the convergence of the EM algorithm. Specifically, they focus on three simple cases where the formulas are tractable, and assess the bias, as well as the convergence properties of the population and sample EM. The three cases considered are instances of Gaussian mixtures: 1. The true model has three classes, with two classes having close means, and the fitting model has only two classes. Variance and weights of the fitted model are fixed according to the true values. 2. The true model has three classes, including one centered in zero and with a small associated weight. The fitting model is the same as above. Variances and weights of the fitting models are fixed according to the true values. 3. Both the true and fitted models have two classes. The weight and variances of the fitted model are fixed, different from the true one. After showing some theoretical results in each situations, the authors validate heuristically their findings with associated simulation studies. # General comments The problem of fitting misspecified statistical models is an important one, and this paper makes a first step toward some theoretical bounds in some specific cases. Although limited and relatively simple, the three cases considered are of practical interest, and give way to consistent developments (although I did not check the proofs in the supplementary material carefully). As a personal opinion, it felt that the paper was missing some context on the state-of-the art. Although some references are given (mainly, Balakrishnan et al. 2017), the results are not compared to the literature, which made it difficult to get an idea of the sharpness of the bounds. In a similar way, I found no reference to the classical model selection theory, which deal with similar problems (trying to find the 'best' number of components to keep in the fitted model). Although the focus is a bit different, it could be interesting to make some parallels. For background on model selection, see e.g.: - Massart, P. (2007). Concentration Inequalities and Model Selection. - Baraud, Y. et al (2009). Gaussian model selection with an unknown variance. AoS. Finally, it felt like the submission could benefit from a proof-reading for punctuation, typos, English style and grammar issues. (See below for some more specific remarks.) # Specific remarks ## Section 2 * Table 1: maybe adding the corresponding section for each model could help the reader (3.1, 3.2 and 4, if I'm correct). (This is just a suggestion.) * It could be useful to add a reference for formulas for section 2.1 (EM for a two-component Gaussian mixture). ## Section 3 * I found the introductory paragraph of section 3 (lines 115 - 124) confusing. It could be worth to refactor it a bit. Specifically: * l.118: "we do not assume any knowledge about the number of component in the true model." Maybe I missed something, but I was under the impression that the true model was assumed to have three or two classes. It might also be in contradiction with the first sentence of the paragraph ("... the number of components in the true model is *larger* than that in the fitted model.") Maybe a clarification could help. * l.119-121: it was not clear to me whether these scenarios were assumptions, or statements of the general nature of misspecification. * l.122-123: "the number of components" is repeated, maybe rephrase. * I felt that a comparison with classical bounds of the EM (that is, when the model is correctly specified) was maybe missing. It could be interesting to recall these bounds, and point out the differences induced by this misspecification. * Similarly, it seems that more comments could be made on the links between the results of sections 3.1 and 3.2. For instance, in seems that the bounds in Th. 1 and Cor. 1 do not depend on the disparity parameter $\rho$, while the ones in Th. 2 and Cor. 2 do depend on $\omega$. Could the authors provide an intuition of why this is the case ? * l.158: should it be $\gamma = e^{-c''\eta^2}$ to match expression l.151 ? * l.177: By multiplying the constant of Prop. 2 by 1/\eta^{1/4}, should the constant $C_\omega = c\omega^{1/8} / \sqrt{1 - \omega}$ ? be (without the $\sigma$) ? ## Section 4 * It could be interesting to comment on the specific form of model (16). In particular, why choosing an expression where both the distance between the two Gaussians and there variance vary conjointly, through parameter $\theta*$ ? * It is not clear from the text whether $\sigma^2$ in the fitted model (17) is fixed or not, and, if so, to which value. ## Simulation studies (section 5) It was not always clear to me how the simulations illustrated the results of the previous sections. Also, the figures are quite small, and I had to magnify Figures 2 and 3 with a factor 3 on my screen to read them properly. I'm myopic, so maybe not representative, but it could ease the reading to leave more space to them, and/or remove some of the panels (see comments below). * I did not understand how the results presented in the third panel of each figure (histogram) was related to the previous theoretical results. Maybe I missed it, but it does not seem to be exploited in the text (beside its description l.248-250). Maybe a clarification could help the reader. * Figure 1: Maybe it could help the reader to give the value of $\bar{\theta}$. From the graph, it seems that it is almost equal to $\theta^*$ (4 or 3), so that there is no bias in this case. On the second panel figure 1b for instance, the sample EM seems to converge to $\theta^* = 3$, while $\rho=0.5$ is large, so the bias should be larger, as hinted by the text (see l.258). Maybe I read the figure wrong, but then a clarification could help. Could a plot of bound $C_\rho$ help ? * Figure 3a: It looks like the fitted model is unimodal (panel 1), while from the text (see Table 1), it should have two components. Again, I might be missing something, but then a clarification could help. ## Discussion Please consider proof reading this discussions, which is difficult to read in its current state because of the many typos (see below). # Possible typos Below are a few possible typos that I noted along the way: * l.52: lower case We ? * l.65: show appears two time, maybe rephrase ? * l.82: should it be "... performance of *the* EM algorithm … " (add 'the') ? Also I think lines 89 ('the location'), 91 ('the EM'), 115 ('the EM'), 133 ('of the EM'), 168 ('the EM'), 176 ('the results'), 177 ('the bias'), 178 ('the EM'a, 270 ('the behavior of the EM')). * l.91: maybe rephrase: "... may not belong to the fit model, …" * l.91: maybe rephrase: "... the best possible estimator that can be achieved using EM can be …" ('can' appears two times). * l.168: missing period at the end of the sentence. * l.270: "general" appears two times. * l.271: "…for the same. illustrate..." maybe a piece is missing ? * l.273: "... is focused … " ? * l.274: "... to ask if we fit a model with scale as well." maybe rephrase ? Also: * "2 or 3 component(s) mixture": this formula appears in several places, with 2 and 3 sometimes spelled out, and with or without an -s at "component". It might be good to unify. * Why is there a lonely section 2.1 ?

Reviewer 3



==== After Author Response ==== The authors acknowledged that the the bounds become loose with ρ, ω and I am satisfied with their response. Regarding the the plots of convergence rate vs ∊ _x000f_I will encourage the authors to include these in the main bodies. Visually the current plots seem to convey duplicate information. the three plots could be condensed into one by just noting in the figure caption that the other plots appear similar. On the other hand, the three different rate of convegence, which can again be plotted on a single axis, will show novel information. Also this way you will save save since you'll have two axes instead of three. ====== END ==== Recommendation: This paper analyses the effect of model-misspecification on the quality and speed of the EM parameter estimation procedure. This analysis is provided for three different types of misspecifications, which are intuitive and plausible. This work is relevant to NIPS, and the numerical simulations corroborates the major claim in the paper that the estimation-error for true parameters decreases as square root of the number of samples. The main contribution of the paper is a thorough and systematic analysis of different types of model-misspecifications which may lead to more robust models and algorithms in future. However, some of the analysis makes some implicit assumptions (See technical issues) and the experimental results could be improved, therefore I have given a conservative rating of 7. Depending on the author response I may update it to 8 or 9. Summary: For each misspecification, the analysis proceeds in three steps. First, the sign-invariant absolute difference between the true parameters and the ideal population-EM estimates is bounded. This difference is called the bias and its upper bound quantifies the worst error that model misspecification can introduce. In the second step, the rate of convergence to the population-EM estimates is established. Depending on the type of mis-specification the exact rate of convergence changes but for all three cases the convergence is linear. Finally, the third step relies upon the machinery developed in [Balakrishnan et al. 2014] to prove that with high-probability the sample-EM iterates converge to a ball around the ideal population-EM estimates. Technical Issues: 1. The upper bounds on the Bias, are derived with few assumptions on hyper-parameters such as ρ and ω but these bounds become useless for some of the values of ρ and ω. For example consider the 3-component mixture with Two-close components. Consider the situation where ρ is 2, in that case the true model actually reduces to a two component model, so the bias should go to zero. But the uppper bound does not reflect that. Similarly with 3-component mixtures with small weight for one component, if ω approaches 1 then the true model actually is just a single gaussian distribution located at origin which can be modelled perfectly by two overlapping gaussians. In this case the bias bound actually reaches infinity. Obviously the upper bounds on bias are still valid, since the true bias is zero but this shows that the quality of the bounds deteriorates with ρ , ω and the analysis is not quite as general and useful as stated in the introduction. 2. Regarding the experiments, since you compute the population EM estimate, why not also show the convergence of the population EM and more importantly check that the convergence rate corresponds to the different linear rates derived for different types of model misspecifications. The statistical error rates for all models is -0.5 which is great but it will be even better to show different convergence rates with different model misspecifications. Although the convergence rate involves unknown constants for models 3.1 and 3.2 it is well-specified as (1-ε²/2) for misspecified variance. Minor Issues and Typos: 118 "we do not assume any knowledge about the number of components in the true model." The analysis depends heavily on the fact that the true model has three components, how is this statement correct? 123 typo, "than the number of components than the actual number of components present in the model." 206 typo "fit the above model -> fit the model below"